# Leukotriene receptor antagonists and eosinophilic granulomatosis with polyangiitis: a disproportionality analysis from FAERS, JADER, CVAR databases integrated with network pharmacology

**Cuilv Liang[1], Pingping Zhuo[2], Yin Zhang**[1]*

**1** Department of Pharmacy, The Second Affiliated Hospital of Fujian Medical University, Quanzhou, China,
**2** Department of Ophthalmology, Fuzhou Changle District People's Hospital, Fuzhou, China

* yizhen6221@163.com

## Abstract

### Objective

The relationship between leukotriene receptor antagonists (LTRAs) usage and the subsequent occurrence of eosinophilic granulomatosis with polyangiitis (EGPA) remained highly polarizing and controversial in previous studies. We aimed to investigate the risk of EGPA caused by LTRAs and the potential toxicological mechanisms of LTRAs-related EGPA.

### Methods

In this real-world pharmacovigilance study, we collected adverse event (AE) reports of EGPA associated with LTRAs use from the U.S. FDA Adverse Event Reporting System (FAERS), Japanese Adverse Drug Event Reporting (JADER), and Canadian Vigilance Adverse Reaction (CVAR) databases. The reporting odds ratio (ROR), proportional reporting ratio (PRR), information component (IC), and empirical Bayesian geometric mean (EBGM) were calculated to quantify the strength of the association between LTRAs and EGPA. The Weibull shape parameter (WSP) test was applied to analyze time-to-onset profiles of EGPA toxicity. Network pharmacology analysis was subsequently performed to identify the central genes to determine the potential mechanisms underlying LTRAs-induced EGPA.

### Results

LTRAs, including montelukast, zafirlukast, and pranlukast, exhibited a strong association with EGPA in three databases (the lower limit of 95% confidence interval (CI) for ROR > 1, PRR > 2 with $\chi^2$ values ≥4, EBGM05 > 2, and IC025 > 0). After excluding corticosteroids as concomitant medication, montelukast remained significantly

**Data availability statement:** Publicly available datasets were analyzed in this study. These data can be found at: FAERS database (https://www.fda.gov/), JADER database (https://www.pmda.go.jp/), CVAR database (https://www.canada.ca/en.html) and GEO database (https://www.ncbi.nlm.nih.gov/geo/).

**Funding:** The author(s) received no specific funding for this work.

**Competing interests:** The authors have declared that no competing interests exist.

**Abbreviations:** EGPA, eosinophilic granulomatosis with polyangiitis; LTRAs, leukotriene receptor antagonists; CysLTs, cysteinyl leukotrienes; AE, adverse event; SRSs, Spontaneous reporting systems; FAERS, U.S. FDA Adverse Event Reporting System; JADER, Japanese Adverse Drug Event Reporting; CVAR, Canadian Vigilance Adverse Reaction; CI, confidence interval; ROR, reporting odds ratio; PRR, proportional reporting ratio; $\chi^2$, chi-square; IC,information component; EBGM, empirical Bayesian geometric mean; TTO, time-to-onset analysis; WSP, Weibull shape parameter; MCODE, molecular complex detection; GO, Gene Ontology; KEGG, Kyoto Encyclopedia of Genes and Genomes.

associated with EGPA in the FAERS database. The median time-to-onset of EGPA associated with LTRAs was 233 (range: 76–660) days, and the WSP test indicated LTRAs had early failure-type profiles. We isolated 81 interactive target genes linking LTRAs to EGPA. Several central genes, including SRC, PTGS2, EDN1, HMOX1, KDR, and OCLN, were revealed via protein-protein interactions analysis and molecular complex detection (MCODE) algorithm.

## Conclusion

Our study revealed LTRAs could increase the risk of EGPA, and initially explored potential genes and mechanisms of LTRAs-induced EGPA. It is helpful for clinicians to be alerted to the risk of EGPA during LTRAs administration.

---

## 1. Introduction

Eosinophilic granulomatosis with polyangiitis (EGPA), formerly known as Churg-Strauss syndrome, is a rare immune-mediated disorder characterized by delayed-onset asthma, eosinophilia in the blood and tissues, and vasculitis of small to medium-sized vessels [1,2]. Global and European pooled estimates of EGPA incidence were 1.22 and 1.07 cases per million person-years, respectively [3]. Although EGPA exhibits a low global and European incidence, it is associated with substantial healthcare resource utilization and disease burden. EGPA is a systemic disease affecting multiple organs and tissues that is difficult to treat and requires long-term follow-up and monitoring [4]. Patients with EGPA experience disease-related risks in the form of high relapse potential, comorbidities, and frequent medical interventions. In recent years, literature has documented a progressive annual increase in the reported occurrence rate of EGPA [5]. The etiology and pathophysiology of EGPA have not been fully elucidated [2]. Several genetic (variants in genes such as HLA-DRB4, GATA3, TSLP, LPP, and BACH2) and environmental (allergens, infections, exposure to silicone substances, etc.) factors may represent triggers of EGPA [6–9]. In addition, evidence suggested that vaccinations and medications (macrolides, thiouracil, hydralazine, minocycline, torsemide, omalizumab, and leukotriene receptor antagonists [LTRAs]) have been shown to potentially induce EGPA [10–13].

Leukotrienes are eicosanoid inflammatory mediators derived from arachidonic acid that play an important role in the inflammatory process in asthma. In the mid-1990s, the FDA (Food and Drug Administration) approved drugs that inhibit the action of leukotrienes by binding to leukotriene receptors, and since then they have been widely used to treat patients with asthma [14]. LTRAs exert their therapeutic effects by blocking the binding of cysteinyl leukotrienes (CysLTs) to their receptors (CysLT1 and G protein-coupled receptor 17), thereby suppressing CysLT-mediated pro-inflammatory signaling [15]. Several LTRAs have been developed, including montelukast, zafirlukast, ibudilast (approved by the FDA), and pranlukast (approved by Japan), which have markedly changed the practice of asthma. LTRAs are relatively well-tolerated and generally safe, they may also cause some adverse events (AEs), among which

EGPA has attracted much attention. The relationship between LTRAs usage and the subsequent occurrence of EGPA remains highly polarizing and controversial. Several studies have reported a temporal association between LTRAs administration and EGPA onset [10,16]. Gramma et al. reviewed the cases of EGPA caused by montelukast reported between 1999 and 2019, identifying montelukast as a frequent trigger of EGPA as an adverse reaction [10]. Paul et al. summarized the literature on the impact of LTRAs treatment on the development of EGPA in asthma patients, which suggested a close relation between LTRAs and the occurrence of EGPA development [16]. However, case reports and case-control studies also supported that LTRAs exhibited an "unmasking" effect, with EGPA observed in asthma patients following reduction/discontinuation of corticosteroids. LTRAs therapy did not increase the risk of EGPA [17–19]. Moreover, most studies have focused on investigating the association between montelukast and EGPA, with limited research on other LTRAs. Thus, the relationship between LTRAs and EGPA is not yet established and requires further investigation.

Since drug-induced EGPA is a rare AE, it is difficult to conduct epidemiologic studies. Spontaneous reporting systems (SRSs), as important pharmacovigilance tools reflecting the realities of clinical practice, are uniquely valuable in rare AEs surveillance [20]. The U.S. FDA Adverse Event Reporting System (FAERS), Japanese Adverse Drug Event Reporting (JADER) database, and Canadian Vigilance Adverse Reaction (CVAR) databases are SRSs containing a large amount of information on drug use and related AEs, and can detect delayed or rare AEs [21]. Disproportionality analysis is the most widespread method for SRSs analysis, which aims to identify and quantify the associations between specific drugs and adverse reactions [22]. In reviewing the published literature, there has been no research that applied disproportionality analysis to explore the association of LTRAs with EGPA in combination with the FAERS, JADER, and CVAR databases. A pharmacovigilance study examining EGPA associated with LTRAs in the FAERS database only included data from September 1997 through April 2003 and did not explore the association of individual LTRA with EGPA [23]. This study aimed to explore the characteristics of EGPA associated with LTRAs by mining three databases and quantifying the strength of the association between individual LTRA and EGPA. The pharmacological mechanisms of EGPA induced by LTRAs remain obscure [24]. Network pharmacology is a rapidly advancing technology that integrates computer science and bioinformatics to explore drug-target interactions and signaling pathways. We employed network pharmacology to elucidate the mechanism of action of LTRAs in EGPA.

## 2. Materials and methods

### 2.1 Data source

We extracted data for the period from January 2004 to March 2025 in the FAERS database, from January 2004 to December 2024 in the JADER database and January 1973 to December 2024 in the CVAR database. The FAERS dataset comprises seven categories of data tables: Demographic Information (DEMO), Drug information (DRUG), Adverse Events (REAC), Outcome (OUTC), Reporting Sources (RPSR), Drug Administration Start/End Dates (THER), and Indications (INDI). The JADER database consists of DEMO, DRUG, REAC, and underlying diseases (HIST). The CVAR database includes 11 categories of data tables. FAERS, JADER, and CVAR are publicly available pharmacovigilance databases containing fully anonymized and de-identified patient records. Therefore, this study did not require ethical approval or informed consent. This study followed the Reporting of a Disproportionality analysis for Drug Safety signal detection using individual case safety reports in PharmacoVigilance (READUS-PV) guidelines (S1 Table). This study complied with the principles of the Declaration of Helsinki. All analyses used publicly available data, therefore did not require informed consent or ethics approval from an institutional review board.

### 2.2 Data extraction

Due to the spontaneous nature of data collection, duplicate reports were presented. For the FAERS database, as recommended by the FDA, we removed duplicate entries in the DEMO table and linked each file via unique

identification number. Regarding the JADER database, duplicate entries were removed from the DRUG and REAC tables, and the DEMO table was subsequently linked to these tables using CASEID [25]. In the CVAR database, after eliminating the duplicate reports, the different data tables were linked using Report_ID [26]. LTRAs considered for selection were montelukast, zafirlukast, pranlukast, and ibudilast. The generic, trade, and former names of LTRAs, which were obtained through the DrugBank database (http://www.drugbank.ca) [27], the PubMed database (https://pubmed.ncbi.nlm.nih.gov/), and the official website of Pharmaceuticals and Medical Devices Agency (https://www.pmda.go.jp/english/), were used for retrieval in three databases (S2 Table). For precise results, only AE reports for drugs in the specified categories in the FAERS database designated as principal suspect and designated as suspect in the JADER and CVAR databases were included. After extracting and standardizing AE reports related to LTRAs, we mapped them to the preferred terms of the Medical Dictionary for Regulatory Activities (https://www.meddra.org/).

## 2.3 Disproportionality analysis

We extracted the clinical characteristics of cases reporting EGPA AEs following the use of LTRAs. These characteristics included age, gender, weight, reporting year, reporter, reporting country, and outcome for descriptive analysis. Disproportionality analysis quantified pharmacovigilance signals based on a 2 × 2 contingency table in SRSs (S3 Table). We jointly applied Bayesian and frequency methods to assess disproportionality between LTRAs and EGPA. The frequency methods comprised the Reporting Odds Ratio (ROR) and Proportional Reporting Ratio (PRR), and Bayesian methods included the Bayesian Confidence Propagation Neural Network (BCPNN) and Multi-Item Gamma Poisson Shrinker (MGPS). Frequency methods demonstrate computational simplicity and high sensitivity, but are prone to false-positive signals when the number of AEs is small. Bayesian methods effectively adjust for uncertainty in limited reporting cases, reducing false-positive signals and exhibiting superior performance in rare AE signal detection [25]. These methods complement each other in the detection of drug safety signals. Consensus across multiple algorithm thresholds enhances the reliability of analytical conclusions. A positive signal was defined as meeting all four methodological criteria: (1) lower limit of the 95% confidence interval (CI) of ROR > 1 and the number of cases ≥3; (2) the 95% CI of PRR > 2 with corresponding chi-square ($\chi^2$) value ≥4; (3) the Information Component 2.5th percentile (IC025) >0; and (4) the lower limit of 95% CI of empirical Bayesian geometric mean (EBGM05) >2 [28,29]. A larger value indicated a stronger signal of an AE, which suggests that the drug has a stronger correlation with the AE. Additionally, to discern the effect of age, gender, indication, and corticosteroid co-administration on EGPA after taking LTRAs, we used the four methods to perform subgroup analyses by age, gender, and indication and co-medication analysis of corticosteroid use. Data were processed and analyzed using Excel 365 (Microsoft Corporation, Redmond, USA) and R software (version 4.2.2; R Foundation for Statistical Computing, Vienna, Austria).

## 2.4 Time-to-onset analysis (TTO)

TTO was defined as the time interval from the initiation of drug therapy to the development of EGPA toxicity. We applied the medians, interquartile range, and Weibull shape parameter (WSP) test to evaluate TTO data. The Weibull distribution test determines the proportional change in the AE rate, indicating the risk of an increase or decrease over time [30]. The Weibull distribution curve is defined by two primary parameters: the scale parameter α (determining the scale of the distribution function) and the shape parameter β (determining the shape of the distribution function). When the shape parameter β was < 1 and its 95% CI was < 1, the hazard was considered to decrease over time (early failure type); when β was equal to or close to 1 and its 95% CI contained the value of 1, the hazard was estimated to remain constant over time (random failure type); when β was > 1 and its 95% CI excluded the value of 1, the hazard was deemed to increase over time (wear failure type) [31].

## 2.5 Network pharmacology analysis: LTRAs-related EGPA network construction

The data required for the comprehensive analysis was obtained from online database platforms [32]. For LTRAs-related targets, including montelukast, zafirlukast, pranlukast, and ibudilast, we searched the SwissTargetPrediction database (https://www.swisstargetprediction.ch/), the Comparative Toxicogenomics database (CTD; https://ctdbase.org/), the Targetnet database (http://targetnet.scbdd.com/home/index/), and the PharmMapper database (https://www.lilab-ecust.cn/pharmmapper/index.html) to predict possible targets. The EGPA-related genes were identified from differentially expressed genes (DEGs) in the row data of GSE144302 (https://www.ncbi.nlm.nih.gov/geo/).

The drug-target list and disease-DEG list were submitted to VENN diagram for obtaining overlapping targets (LTRAs-EGPA genes). The overlapping targets of LTRAs and EGPA were the potential targets that could regulate EGPA. The overlapping targets were investigated by the STRING website (https://string-db.org/), with a focus on Homo sapiens species with a 0.4 confidence threshold. Then, Cytoscape software was employed to visualize the protein-protein interaction network. In this network, biological entities (LTRAs-EGPA-related genes) were depicted as nodes, and the interactions between them were represented as edges. The molecular complex detection (MCODE) algorithm was applied to screen for key cluster [33].

The LTRAs-EGPA genes were analyzed by Gene Ontology (GO) and Kyoto Encyclopedia of Genes and Genomes (KEGG) for functional enrichment [34]. Through GO analysis of these genes, we identified three categories of cellular components, biological processes, and molecular functions to investigate the biological characteristics of LTRAs-associated EGPA genes. Meanwhile, KEGG enrichment analysis was performed to predict potential signaling pathways that may be implicated in LTRAs-induced EGPA.

## 3. Results

### 3.1 Basic characteristics of AE reports

The FAERS, JADER, and CVAR databases recorded 104,037, 1,948, 37,404 LTRAs-related AE cases and 2,509, 466, 237 EGPA-related cases, respectively. These cases included 1,031 LTRAs-related EGPA (FAERS: 822, JADER: 150, CVAR: 61) reports involving montelukast (949 cases, FAERS: 817, JADER: 71, CVAR: 61), pranlukast (75 cases, JADER: 75), and zafirlukast (9 cases, FAERS: 5, JADER: 4). No reports of EGPA with ibudilast coded as the "principal suspect" drug in the FAERS database or as a "suspect" drug in the JADER and CVAR databases were identified (Fig 1). Fig 2 presented the characteristics of patients experiencing EGPA AEs associated with LTRAs in the three databases. The highest numbers of LTRAs associated with EGPA AEs were reported before 2004 and after 2024, whereas the number of reported cases remained stable in other years (Fig 2a). A higher proportion of EGPA AEs were reported in females in the FAERS and JADER databases, whereas males predominated in the CVAR database (Fig 2b). The data from the three databases revealed that the highest proportion of cases was observed in the 18–65 years subgroup, followed by the ≥ 65 years subgroup (Fig 2c). Patients with weight ≥50 kg exhibited a higher proportion of EGPA cases compared to those <50 kg (Fig 2d). Reports from healthcare professionals accounted for a greater proportion of EGPA cases compared to non-healthcare professionals (Fig 2e).
For montelukast, pranlukast, and zafirlukast, the majority of EGPA cases were concentrated in individuals aged 18–65 years and were reported by physicians (Figs 2c and 2e). For montelukast, most EGPA reports originated from the United States, followed by the United Kingdom, Japan, and Canada (Fig 2f). Among the 878 case reports that documented outcomes of montelukast-associated EGPA, severe clinical outcomes were observed in 415 patients, with hospitalization being the most frequently reported outcome (Fig 2g). For pranlukast, all EGPA reports were derived from Japan. Clinical outcomes were documented in 58 pranlukast-related EGPA, of whom 23 recovered, while 16 experienced unrecovered or sequelae (Fig 2h).

### 3.2 Disproportionality analysis

ROR, PRR, EBGM, and IC for LTRAs-associated EGPA AEs were calculated for all databases. Data mining algorithms (the lower limit of 95% CI for ROR > 1, PRR > 2 with $\chi^2$ values ≥4, EBGM05 > 2, and IC025 > 0) demonstrated that the

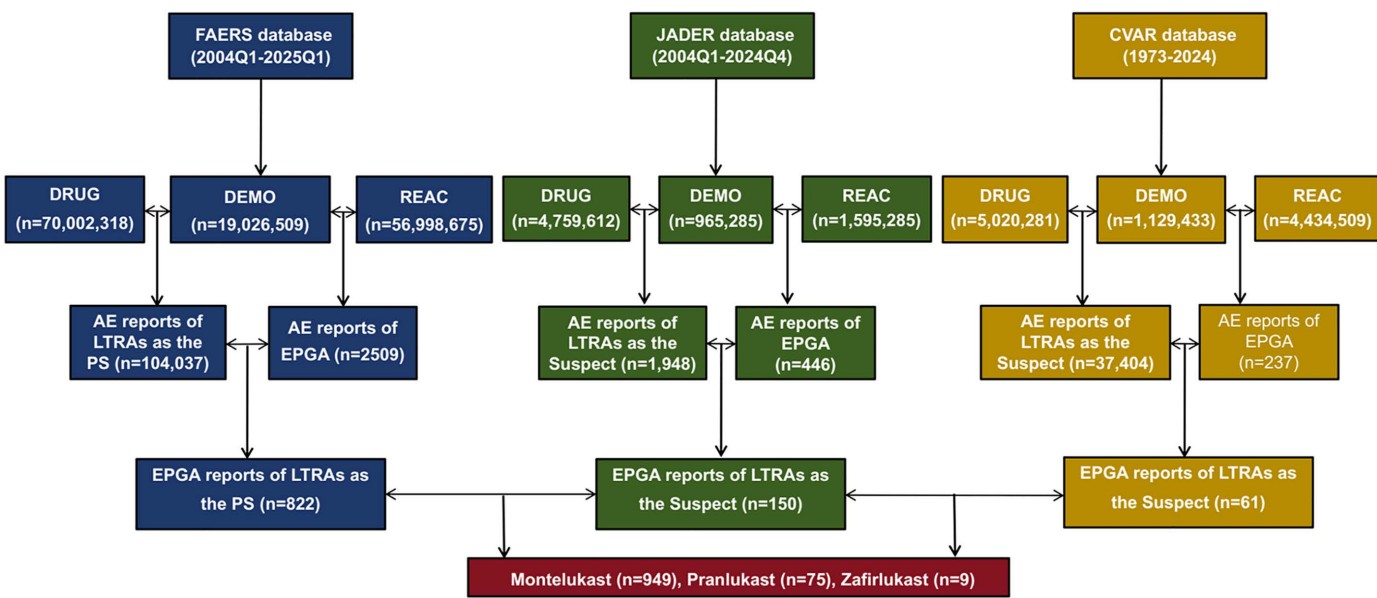

**Fig 1. Flowchart of EGPA adverse event reports identified from the FAERS, JADER, and CVAR databases with LTRAs.** FAERS, Food and Drug Administration Adverse Event Reporting System; JADER, Japanese Adverse Drug Event Reporting; CVAR, Canadian Vigilance Adverse Reaction; LTRAs, leukotriene receptor antagonists; EGPA, eosinophilic granulomatosis with polyangiitis; DEMO, demographic information table; DRUG, drug information table; AE, adverse event; REAC, adverse events table; PS, primary suspect drug.

LTRAs exhibited a strong association with EGPA in three databases. Individual LTRA agents, including montelukast, pranlukast, and zafirlukast, also demonstrated strong correlations with EGPA (Table 1).

The descriptive values of the subgroup analysis determined whether montelukast/pranlukast were associated with EGPA (Table 2). The reports were gathered based on patient demographics (gender, age, and indication). Age was stratified by placing the study population into two distinct groups: <65 and ≥65 years. For montelukast/pranlukast-associated EGPA, the age subset analysis demonstrated that compared to the <65 years, the ≥65 years had higher signal strength. Among the indication-based groups, which were stratified into asthma and other indications (e.g., rhinitis, hypersensitivity, hypertension, chronic obstructive pulmonary disease), other indications demonstrated a higher signal strength than asthma for montelukast/pranlukast-associated EGPA. For montelukast-associated EGPA, the gender subset analysis revealed that females had a stronger signal intensity than males in the FAERS and JADER databases, while the signal intensity was stronger for males in the CVAR database. The gender subset analysis demonstrated that males had the greater signal strength for pranlukast-associated EGPA.

The top 10 medications most frequently co-administered with montelukast were selected from three databases. As shown in Fig 3, prednisone, fluticasone, budesonide, and salmeterol were consistently reported across all three databases. Among montelukast-associated EGPA AEs documented in the FAERS database, 64, 198, and 94 cases involved concomitant use with prednisone, fluticasone, and budesonide, respectively. The combination with corticosteroid did not enhance the signal strength of EGPA associated with montelukast (Table 3). After sequential exclusion of these concomitant medications, disproportionality analysis revealed that montelukast maintained a statistically significant correlation with EGPA (the lower limit of 95% CI for ROR > 1, PRR > 2 with $\chi^2$ values ≥4, EBGM05 > 2, and IC025 > 0).

### 3.3 TTO

Among the EGPA toxicity AE reports associated with LTRAs, including montelukast, pranlukast, and zafirlukast, time-to-onset data were available for 316 reports. The median time-to-onset of EGPA associated with LTRAs was 233 (range:

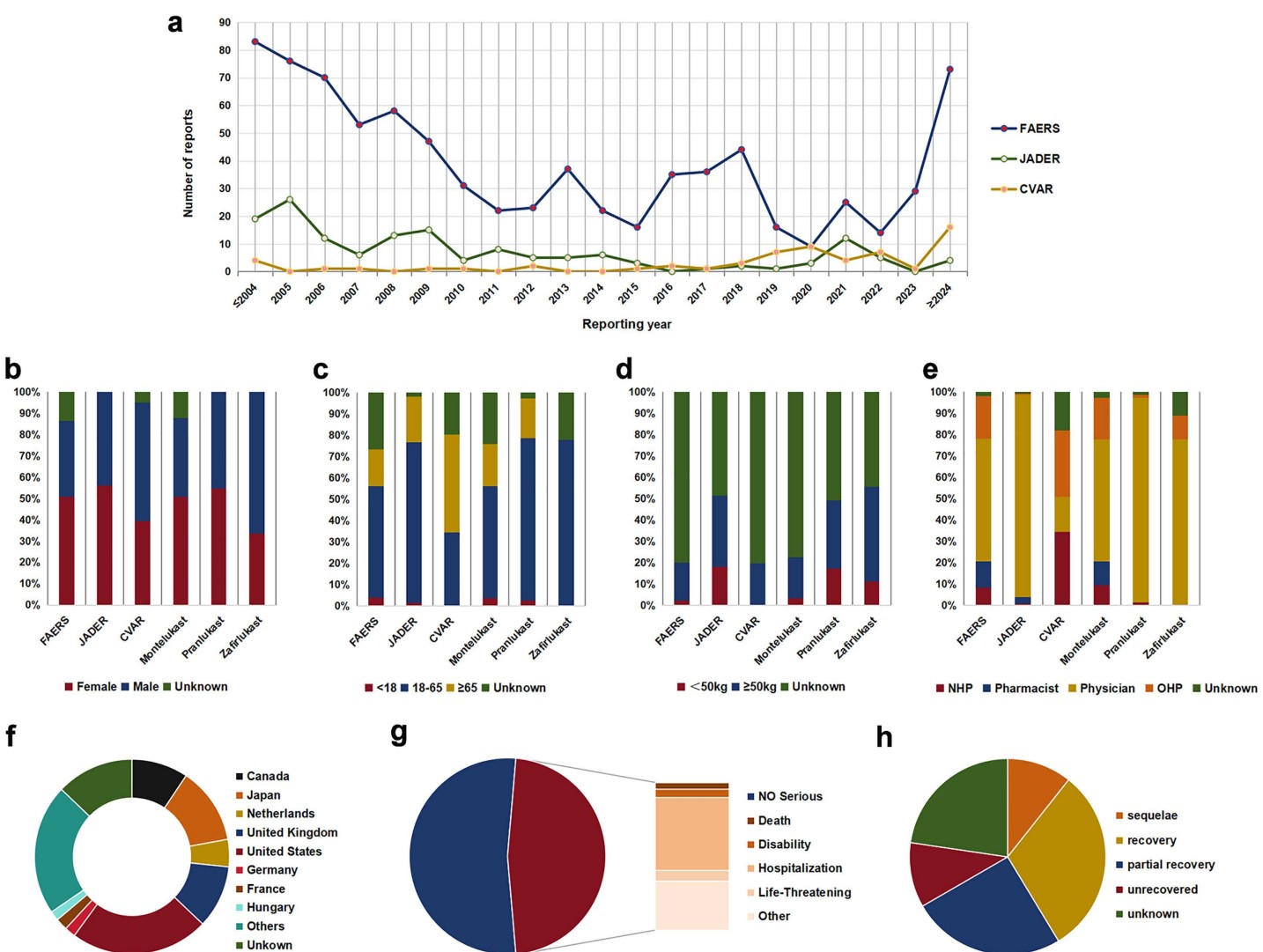

**Fig 2. Basic information of LTRAs associated with EGPA toxicity. (a)** The number of annual adverse reaction reports. **(b)** The gender distribution of the patients. **(c)** The age distribution of the patients. **(d)** The weight distribution of the patients. **(e)** The occupational distribution of the reporter. **(f)** The top 8 countries with the largest number of montelukast-related EGPA reports. **(g)** The outcome distribution of montelukast-related EGPA in patients. **(h)** The outcome distribution of pranlukast-related EGPA in patients. Note: Fig 2 contains only statistical data visualizations and does not include any patient imaging or re-imaging data. LTRAs, leukotriene receptor antagonists; EGPA, eosinophilic granulomatosis with polyangiitis; FAERS, Food and Drug Administration Adverse Event Reporting System; JADER, Japanese Adverse Drug Event Reporting; CVAR, Canadian Vigilance Adverse Reaction; NHP, non-health professional; OHP, other health professional.

76–660) days. In the WSP test, the lower limit of 95% CI of the shape parameter was < 1, suggesting an early failure type curve profile. For each LTRA, time-to-onset data were available for 284 EGPA-toxicity AE reports associated with montelukast and 27 associated with pranlukast. The median time-to-onset of EGPA associated with montelukast and pranlukast was 244 days (range: 76–679) and 150 days (range: 44–292), respectively. In the WSP tests for montelukast and pranlukast, the lower limit of 95% CI for the shape parameter was also <1, suggesting an early failure type curve. Results of the WSP test and histograms of the incidence were displayed in Table 4 and Fig 4.

**Table 1. Results of disproportionality analysis of LTRAs-associated EGPA.**

| Database | Drugs | n | ROR (95%CI) | PRR (χ²) | EBGM (EBGM05) | IC (IC025) |
|---|---|---|---|---|---|---|
| FAERS | LTRAs | 822 | 268.58 (247.04-292.00) | 266.46 (146177.12) | 179.49 (167.37) | 7.49 (7.37) |
| | Montelukast | 817 | 269.07 (247.46-292.57) | 266.94 (145981.97) | 180.34 (168.14) | 7.49 (7.38) |
| | Zafirlukast | 5 | 102.07 (42.36-245.93) | 101.62 (497.17) | 101.42 (48.59) | 6.66 (5.48) |
| JADER | LTRAs | 150 | 449.22 (367.12-549.67) | 414.70 (41096.89) | 275.56 (232.75) | 8.11 (6.43) |
| | Montelukast | 71 | 280.44 (216.10-363.94) | 263.14 (15594.04) | 221.41 (178.03) | 7.79 (6.12) |
| | Zafirlukast | 4 | 465.57 (163.66-1324.43) | 412.48 (1627.64) | 408.79 (170.44) | 8.68 (6.94) |
| | Pranlukast | 75 | 545.33 (420.09-707.90) | 484.03 (30081.22) | 402.80 (323.80) | 8.65 (6.98) |
| CVAR | Montelukast | 61 | 40.81 (30.49-54.61) | 40.74 (1756.30) | 30.51 (23.91) | 4.93 (3.26) |

FAERS, Food and Drug Administration Adverse Event Reporting System; JADER, Japanese Adverse Drug Event Reporting; CVAR, Canadian Vigilance Adverse Reaction; LTRAs, leukotriene receptor antagonists; EGPA, eosinophilic granulomatosis with polyangiitis; ROR, reporting odds ratio; CI, confidence interval; PRR, proportional reporting ratio; $\chi^2$, chi-squared; EBGM, Empirical Bayes Geometric Mean; EBGM05, lower limit of 95% CI of EBGM; IC, information component; IC025: Information Component 2.5th percentile.

### 3.4 LTRAs-related EGFA network construction

After deduplication of the databases, we identified 553 target genes linked to LTRAs, along with 3,555 genes linked to EGPA (S4 Table). By intersecting these genes, we separated 81 interacting target genes that represented the intersection of LTRAs target genes and EGPA-related genes (S1 Fig). We conducted protein-protein interaction predictions for these intersecting genes via the String database (Fig 5a). Upon the MCODE algorithm, a key cluster of the interaction was identified, including SRC, PTGS2, EDN1, HMOX1, KDR, and OCLN, revealing their centrality within the network (Fig 5b).

KEGG analysis of LTRAs-EGPA genes was shown in Fig 5c. We focused on the top 20 pathways for comprehensive mapping. The analysis revealed that genes interacting with LTRAs-EGPA were enriched in different pathways, particularly VEGF signaling pathway, renin secretion, and drug metabolism. These findings suggested that LTRAs may be involved in the development of EGPA by regulating these pathways. The GO analysis results of the LTRAs-EGPA genes were presented in S2 Fig. The biological process category of GO exhibited enrichment in vascular processes in circulatory system and muscular systems. The cellular component category was enriched in membrane raft and membrane microdomain. The molecular function category showed enrichment in amide binding oxidoreductase activity and acting on the CH-OH group of donors, NAD or NADP as acceptor.

### 4. Discussion

EGPA is a rare yet resource-intensive systemic immune-mediated disorder that can be triggered by a variety of factors, among which drug-induced cases warrant particular attention. Although many studies have investigated the potential association between LTRAs and the risk of EGPA, existing findings remain controversial. Notably, the majority of reported cases have focused exclusively on montelukast, with little attention paid to other LTRAs. Our study is the first to employ disproportionality analysis to comprehensively reveal the correlation between several LTRAs and EGPA risk by combined with three large-scale datasets from the FAERS, JADER, and CVAR databases. Four algorithms consistently demonstrated that LTRAs, including montelukast, zafirlukast, and pranlukast, exhibited a strong association with EGPA in three databases. Individual LTRA agents, including montelukast, pranlukast, and zafirlukast, also demonstrated strong links with EGPA. Furthermore, after excluding corticosteroids as concomitant medication, montelukast maintained significantly associated with EGPA. To better understand the toxicological mechanisms of LTRAs-associated EGPA AEs, we curated drug-gene interactions from public databases. By searching for key cluster of LTRAs-associated EGPA genes, we finally identified these key genes, including SRC, PTGS2, EDN1, HMOX1, KDR, and OCLN, suggesting that they play important roles in mediating the interaction between LTRAs and EGPA pathways.

                                                                                  

**Table 2. Results of subgroup disproportionality analysis of montelukast/pranlukast-associated EGPA based on gender, age, and indication.**

**Montelukast**

**A. Gender**

| Database | | n | ROR (95%CI) | PRR ($\chi^2$) | EBGM (EBGM05) | IC (IC025) |
|---|---|---|---|---|---|---|
| FAERS | Female | 415 | 397.26 (350.21-450.63) | 394.40 (95269.10) | 231.14 (208.00) | 7.85 (7.68) |
| | Male | 290 | 216.60 (188.79-248.51) | 214.93 (43554.00) | 151.88 (135.38) | 7.25 (7.05) |
| JADER | Female | 42 | 340.17 (240.18-481.79) | 316.66 (10575.71) | 253.53 (189.47) | 7.99 (6.30) |
| | Male | 29 | 303.06 (201.73-455.28) | 285.75 (6919.29) | 240.38 (171.00) | 7.91 (6.22) |
| CVAR | Female | 24 | 31.89 (20.22-50.31) | 31.87 (553.54) | 24.81 (16.95) | 4.63 (2.94) |
| | Male | 34 | 49.27 (33.23-73.05) | 49.13 (1170.51) | 36.14 (25.99) | 5.18 (3.49) |

**B. Age**

| Database | | n | ROR (95%CI) | PRR ($\chi^2$) | EBGM (EBGM05) | IC (IC025) |
|---|---|---|---|---|---|---|
| FAERS | <65 | 459 | 264.26 (234.23-298.14) | 262.45 (69025.16) | 151.95 (137.36) | 7.25 (7.09) |
| | ≥65 | 142 | 419.88 (346.24-509.18) | 414.43 (43022.65) | 304.70 (259.29) | 8.25 (7.98) |
| JADER | <65 | 52 | 343.80 (251.43-470.12) | 313.44 (13191.79) | 255.41 (196.58) | 8.00 (6.32) |
| | ≥65 | 18 | 560.91 (328.48-957.79) | 529.11 (7406.34) | 413.19 (264.06) | 8.69 (6.99) |
| CVAR | <65 | 21 | 25.71 (15.97-41.38) | 25.68 (402.23) | 20.93 (14.05) | 4.39 (2.70) |
| | ≥65 | 28 | 56.45 (36.03-88.44) | 56.29 (1036.79) | 38.70 (26.58) | 5.27 (3.58) |

**C. Indication**

| Database | | n | ROR (95%CI) | PRR ($\chi^2$) | EBGM (EBGM05) | IC (IC025) |
|---|---|---|---|---|---|---|
| FAERS | Asthma | 566 | 14.11 (12.66-15.72) | 13.97 (3961.59) | 8.53 (7.79) | 3.09 (2.95) |
| | Others | 251 | 328.87 (285.90-378.31) | 327.13 (63950.96) | 256.56 (228.19) | 8.00 (7.80) |
| JADER | Asthma | 53 | 12.23 (8.92-16.75) | 11.19 (393.42) | 9.07 (6.97) | 3.18 (1.50) |
| | Others | 8 | 246.26 (119.07-509.32) | 240.79 (1774.08) | 223.66 (121.76) | 7.81 (6.11) |

**Pranlukast**

**A. Gender**

| Database | | n | ROR (95%CI) | PRR ($\chi^2$) | EBGM (EBGM05) | IC (IC025) |
|---|---|---|---|---|---|---|
| JADER | Female | 41 | 544.53 (380.96-778.31) | 486.04 (15993.74) | 391.79 (290.57) | 8.61 (6.93) |
| | Male | 34 | 715.36 (482.78-1059.99) | 628.31 (17320.5) | 511.12 (367.81) | 9.00 (7.31) |

**B. Age**

| Database | | n | ROR (95%CI) | PRR ($\chi^2$) | EBGM (EBGM05) | IC (IC025) |
|---|---|---|---|---|---|---|
| JADER | <65 | 59 | 731.74 (537.18-996.76) | 609.26 (28315.51) | 481.55 (371.81) | 8.91 (7.23) |
| | ≥65 | 14 | 1204.01 (656.1-2209.47) | 1058.82 (12271.13) | 878.22 (528.43) | 9.78 (8.06) |

**C. Indication**

| Database | | n | ROR (95%CI) | PRR ($\chi^2$) | EBGM (EBGM05) | IC (IC025) |
|---|---|---|---|---|---|---|
| JADER | Asthma | 64 | 37.12 (27.16-50.74) | 29.01 (1309.69) | 21.98 (16.92) | 4.46 (2.77) |
| | Others | 6 | 243.71 (106.12-559.69) | 238.26 (1341.77) | 225.55 (112.49) | 7.82 (6.12) |

FAERS, Food and Drug Administration Adverse Event Reporting System; JADER, Japanese Adverse Drug Event Reporting; CVAR, Canadian Vigilance Adverse Reaction; EGPA, eosinophilic granulomatosis with polyangiitis; ROR, reporting odds ratio; CI, confidence interval; PRR, proportional reporting ratio; $\chi^2$, chi-squared; EBGM, Empirical Bayes Geometric Mean; EBGM05, lower limit of 95% CI of EBGM; IC, information component; IC025: Information Component 2.5th percentile.

Following the introduction of LTRAs into asthma management, the association between LTRAs and EGPA was reported [14,18]. As LTRAs have gradually become an integral component of asthma treatment, increasing attention has been directed toward determining whether a genuine correlation exist between LTRAs and EGPA. Consistent with the majority of previous findings, our study demonstrated that LTRAs exhibited significant signs of EGPA toxicity, irrespective of

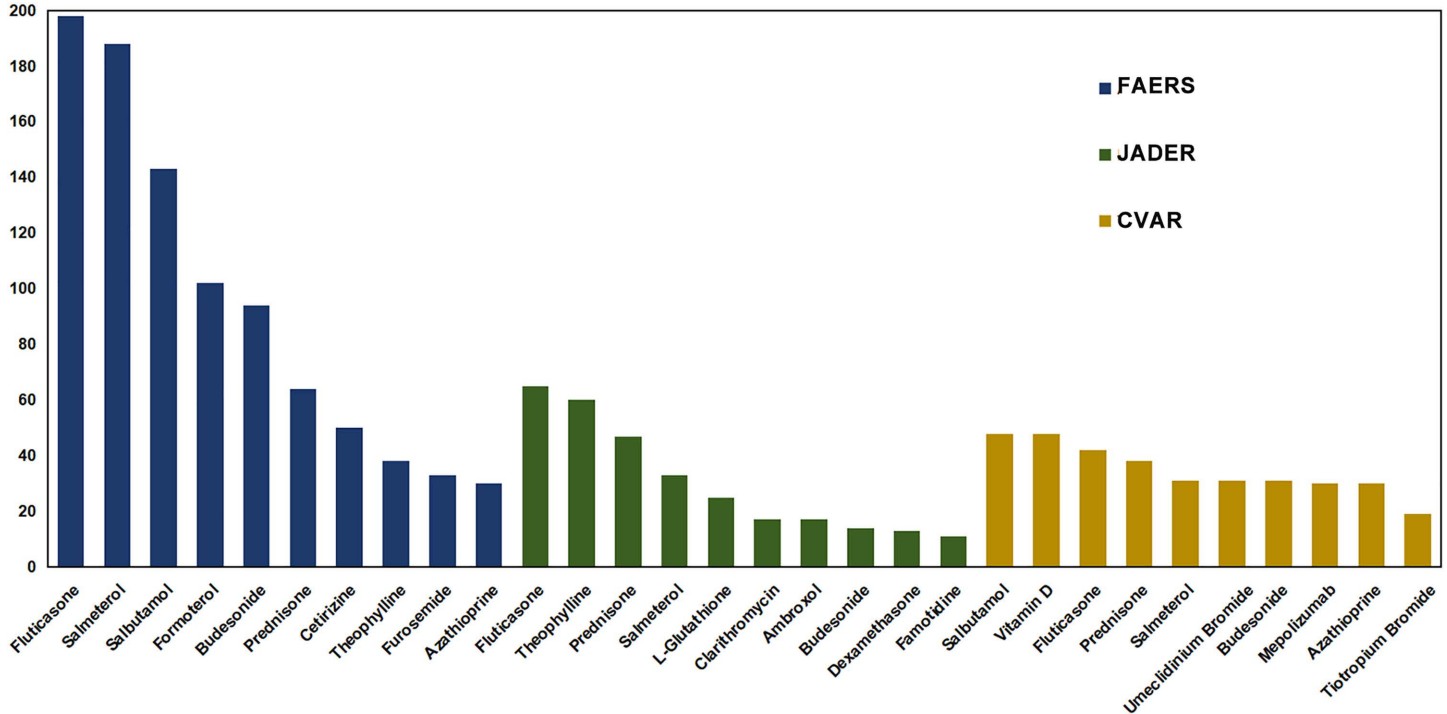

**Fig 3. The top 10 medications most frequently co-administered with montelukast-associated EGPA in the FAERS, JADER, and CVAR databases.** FAERS, Food and Drug Administration Adverse Event Reporting System; JADER, Japanese Adverse Drug Event Reporting; CVAR, Canadian Vigilance Adverse Reaction; EGPA, eosinophilic granulomatosis with polyangiitis.

**Table 3. Results of co-medication disproportionality analysis of montelukast-associated EGPA in the FAERS database.**

**A. Concomitant drugs**

|  | n | ROR (95%CI) | PRR ($\chi^2$) | EBGM (EBGM05) | IC (IC025) |
|---|---|---|---|---|---|
| Prednisone | 64 | 140.53 (109.56-180.25) | 139.69 (8587.25) | 136.14 (106.14) | 7.09 (5.10) |
| Fluticasone | 198 | 259.41 (224.20-300.15) | 256.73 (46440.73) | 236.46 (204.36) | 7.89 (6.55) |
| Budesonide | 94 | 161.65 (131.46-198.78) | 160.56 (14344.82) | 154.55 (125.69) | 7.27 (5.58) |
| B. Removal of cases with concomitant drugs | | | | | |
|  | n | ROR (95%CI) | PRR ($\chi^2$) | EBGM (EBGM05) | IC (IC025) |
| Prednisone removed | 753 | 267.52 (245.55-291.46) | 265.34 (138530.43) | 185.66 (170.41) | 7.54 (7.10) |
| Fluticasone removed | 619 | 224.42 (204.88-245.82) | 222.77 (102800.01) | 167.82 (153.21) | 7.39 (6.92) |
| Budesonide removed | 723 | 261.42 (239.71-285.11) | 259.31 (132197.52) | 184.55 (169.22) | 7.53 (7.08) |

FAERS, Food and Drug Administration Adverse Event Reporting System; EGPA, eosinophilic granulomatosis with polyangiitis; ROR, reporting odds ratio; CI, confidence interval; PRR, proportional reporting ratio; $\chi^2$, chi-squared; EBGM, Empirical Bayes Geometric Mean; EBGM05, lower limit of 95% CI of EBGM; IC, information component; IC025: Information Component 2.5th percentile

the specific LTRA administered. Nathani et al. identified 62 cases of EGPA following LTRAs therapy through systematic literature retrieval (montelukast:29, zafirlukast:17, pranlukast:16). Evaluation based on the Hill criteria for strength of causality suggested that a potential causal relationship between LTRAs and EGPA [35]. A case-crossover study revealed a 4.5-fold increased risk of EGPA onset within the first three months of montelukast treatment [36]. However, both studies emphasized the rarity of EGPA, making it difficult to detect significant associations in statistical tests. Subsequently,

**Table 4. Weibull shape parameter test for EGPA associated with LTRAs.**

|  | Case reports | Median (d) (25%−75%) | Scale parameter: α (95% CI) | Shape parameter: β (95% CI) | Type |
|---|---|---|---|---|---|
| LTRAs | 316 | 233 (76-660) | 391.08 (331.19-450.97) | 0.76 (0.69-0.82) | early failure |
| Montelukast | 284 | 244 (76-679) | 407.26 (341.82-472.71) | 0.76 (0.69-0.83) | early failure |
| Pranlukast | 27 | 150 (44-292) | 254.86 (120.84-388.88) | 0.75 (0.53-0.98) | early failure |

CI, confidence interval; d, days; LTRAs, leukotriene receptor antagonists; EGPA, eosinophilic granulomatosis with polyangiitis.

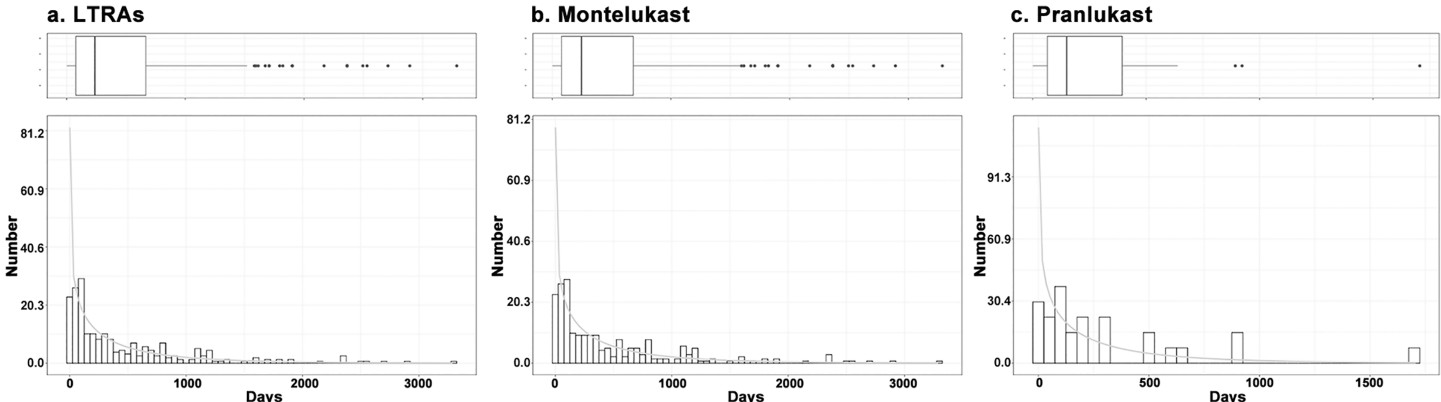

**Fig 4. Histogram of EGPA associated with LTRAs.** (a) LTRAs, including montelukast, pranlukast, and zafirlukast; (b) montelukast; (c) pranlukast. LTRAs, leukotriene receptor antagonists; EGPA, eosinophilic granulomatosis with polyangiitis.

several cases of EGPA induced by montelukast were reported [24,37,38]. At the same time, the EGPA had progressed not only with montelukast but also with zafirlukast and pranlukast [16]. Two recent systematic studies of cases also reported a potential causal link between LTRAs and EGPA [10,16]. The summary of product characteristics for montelukast, zafirlukast, and pranlukast specifically highlighted that LTRAs may cause eosinophilia, particularly Churg-Strauss syndrome. Therefore, our study considered that EGPA may be associated with all LTRA-class medications.

There have been some case reports supporting the absence of a direct causal relationship between montelukast and EGPA, as EGPA was observed in asthma patients only after reducing or discontinuing oral corticosteroids [17–19]. To elucidate the independent effects of LTRAs, Guisti Del Giardino et al. compared the characteristics of patients with and without LTRAs exposure. None of the patients in the LTRAs-exposed group had a history of oral corticosteroid use, suggesting that steroid withdrawal was not a critical factor in the development of EGPA [39]. Among the 8 patients treated with pranlukast reported by Shimbo et al., 3 had never received corticosteroid therapy before the onset of EGPA [40]. Approximately 30%−40% of EGPA patients tested positive for anti-neutrophil cytoplasmic antibodies, and a single-center retrospective study by Schroeder et al. demonstrated that a significant correlation between LTRAs exposure and anti-neutrophil cytoplasmic antibodies positivity in EGPA patients [41]. Alexander et al. performed a literature review of cases of EGPA treated with montelukast in patients who had not previously used oral corticosteroids, supplemented by two self-reported cases, finding a clear causal relationship between montelukast and the occurrence of systemic EGPA with eosinophilia [42]. In our study, montelukast remained significantly correlated with EGPA regardless of concomitant corticosteroids use or exclusion. Regrettably, due to the limited number of EGPA-related AEs reported for pranlukast and zafirlukast, the impact of corticosteroids could not be further evaluated. In subsequent studies, we will progressively follow up on relevant pharmacovigilance data to clarify the effects.

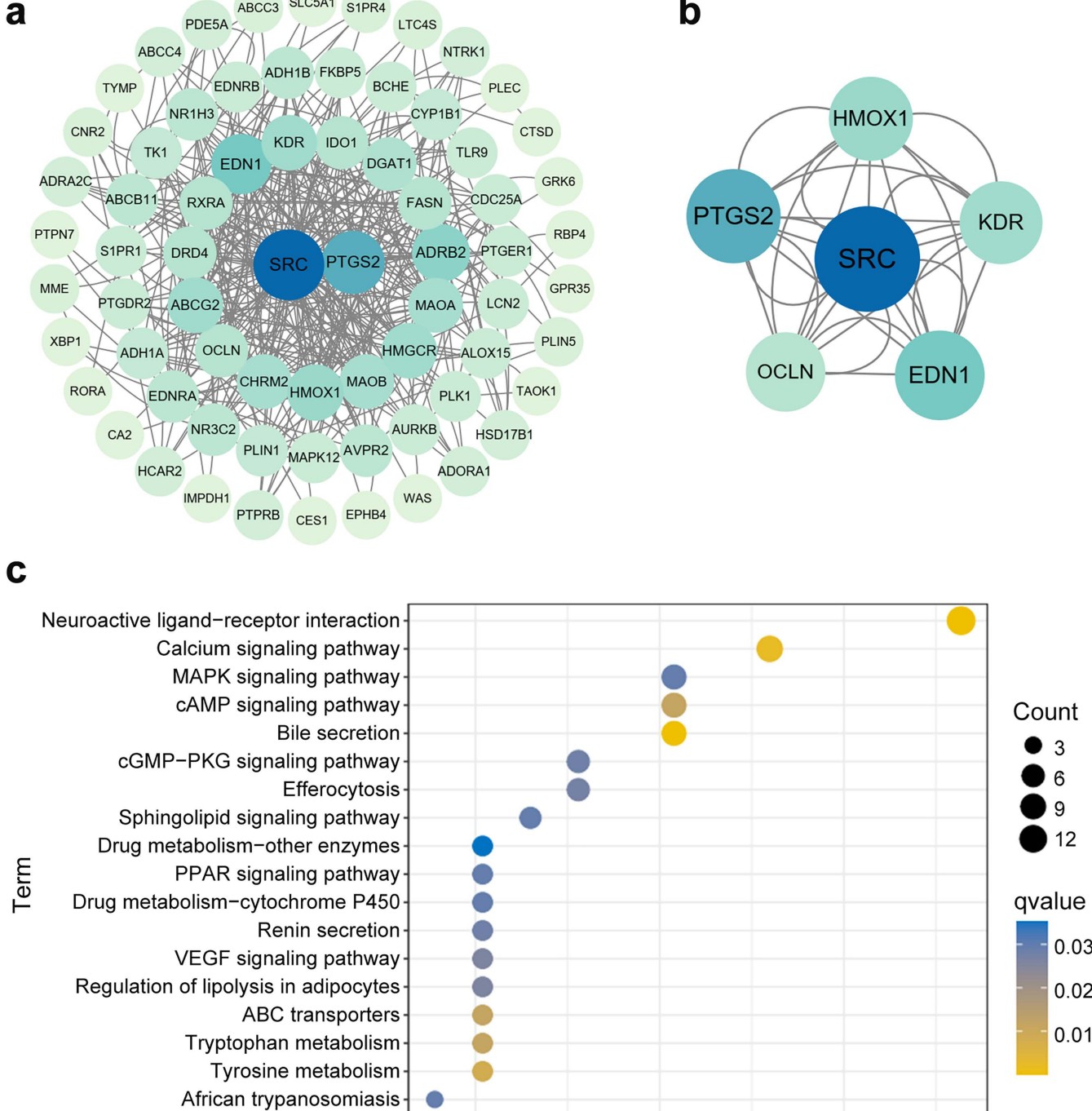

**Fig 5. Results of gene interaction network analysis of LTRAs-associated EGPA.** Protein-protein interaction network of LTRAs-associated EGPA genes presented by Cytoscape. (b) Key cluster genes based on the MCODE algorithm. (c) KEGG pathway enrichment analysis of LTRAs-associated EGPA genes LTRAs, leukotriene receptor antagonists; EGPA, eosinophilic granulomatosis with polyangiitis; MCODE, molecular complex detection; KEGG, Kyoto Encyclopedia of Genes and Genomes.

Meanwhile, subgroup analyses of montelukast and pranlukast users indicated that EGPA toxicity exerted significant effects across all age groups, genders, and indications, particularly in patients aged ≥65 years and those with non-asthma indications. These patients often lack systematic monitoring and had comorbidities, which can readily mask EGPA symptoms. Heightened vigilance for EGPA toxicity is warranted. TTO results indicated that the median time to onset for montelukast and pranlukast was 244 days and 150 days, respectively. The study by Alexander et al. also reported that the majority of patients (18/26) developed EGPA within 1 year of starting montelukast therapy [42]. In the 8 patients reported by Shimbo et al., the onset of EGPA occurred within 1 and 6 months after the initiation of pranlukast administration [40]. Moreover, WSP analysis revealed that LTRAs-related EGPA events were characterized by an early failure type, suggesting that the risk progressively decreases over time. The initial phase of LTRAs treatment represents a high-risk window for the occurrence of EGPA, necessitating proactive screening, intensified monitoring, and preemptive rapid contingency protocols.

The network pharmacology approach indicated important insights into the potential mechanisms of LTRAs-induced EGPA. The protein encodes by the PTGS2 gene, cyclooxygenase-2, which catalyzes prostaglandin E2 synthesis, mediates inflammatory responses, and regulates vascular permeability [43]. KDR encodes vascular endothelial growth factor receptor 2, playing a pivotal role in angiogenesis, vascular development, and vascular permeability [44]. SRC kinase is extensively involved in biological processes such as cell proliferation, cytoskeletal reorganization, and OCLN is a critical component of tight junctions, maintaining epithelial barrier function [45,46]. EDN1 encodes endothelin-1, a potent vasoconstrictor that has a strong effect on vascular tone and vascular smooth muscle cell proliferation, while HMOX1 regulates vascular cell proliferation in a cell-specific manner [47,48]. LTRAs suppress inflammation by blocking leukotriene activity. Leukotrienes, prostaglandins, and other inflammatory mediators interact intricately within inflammatory networks [49]. EGPA is also a vasculitis characterized by damage and destruction of blood vessels and tissue barriers, as well as abnormal angiogenesis. LTRAs exert their effects by targeting CysLT1 receptors in pulmonary tissues, blocking the actions of leukotriene C4 and D4. This leads to an overall increase in circulating leukotrienes, which subsequently act on uninhibited CysLT2 receptors predominantly distributed within the vascular system [50]. The impact of LTRAs on inflammatory mediators or cytokines may indirectly modulate the vascular endothelial microenvironment and inflammation-associated pathways mediated by genes such as SRC, PTGS2, EDN1, HMOX1, KDR, and OCLN, ultimately influencing the pathophysiological progression of EGPA [51–53]. In our study, KEGG analysis demonstrated that LTRAs-EGPA genes were enriched in the VEGF and drug metabolism signaling pathways, and the biological process category of GO exhibited enrichment in vascular processes in circulatory system. Through these key genes, we conjectured that the vascular endothelial microenvironment, VEGF pathway, and inflammation may be the possible mechanisms of EGPA-induced LTRAs.

Our study has several limitations. First, although the three databases encompass millions of AE reports, they are SRSs with inherent limitations, which do not provide exact records of drug dosage changes or tapering regimens, prior medication history, disease severity, detailed comorbidity data, steroid intolerance or comorbid allergic conditions of patients in a structured or comprehensive manner. Second, the anonymous, voluntary nature of SRSs makes them susceptible to under-reporting, duplicate reports, incomplete data, and reporting bias. To mitigate these limitations, we utilized three large-sample databases and four data mining algorithms for cross-validation. Disproportionality analyses were restricted to AE reports in which LTRAs were coded as "principal suspect" or "suspect" drugs to minimize dilution effects from concomitant medications. A co-medication analysis of corticosteroid use was performed to minimize confounding by steroid therapy. Analyses stratified by age, gender, and indication were conducted to demonstrate that the findings were not driven by a single demographic characteristic or indication cluster. Despite our attempts to mitigate these biases, residual confounding cannot be fully eliminated. The observed association may still overestimate the true causal effect of LTRAs on EGPA. Disproportionality analysis can also only assess signal strength, neither establishing causality nor quantifying real-world risks. Therefore, the findings of this study cannot establish causality or quantify the independent contribution of

LTRAs. Further validation is required through longitudinal prescription databases or prospective pharmacovigilance cohort studies. Third, this study preliminarily explored the potential targets and mechanisms of the development of EGPA caused by LTRAs using network pharmacology approach. Further validation through cellular or animal experiments is required to elucidate the actual roles of vascular endothelial microenvironment, the VEGF pathway, and inflammatory responses in LTRAs-induced EGPA.

## 5. Conclusion

Given the rapid increase in LTRAs use worldwide, EGPA needs to be examined in multiple large spontaneous AE reporting databases to understand the relationship between LTRAs use and EGPA AEs. Our study applied real-world data from the FAERS, JADER, and CVAR databases to measure the risk of EGPA among several LTRAs through time of onset and disproportionality analysis. LTRAs, including montelukast, zafirlukast, and pranlukast, were found to correlate with EGPA toxicity. Individual LTRA agents, including montelukast, pranlukast and zafirlukast, showed the same significant association. After excluding corticosteroids as concomitant medication, montelukast remained significantly associated with EGPA. Potential mechanisms by which LTRAs contribute to EGPA were revealed by pharmacogenetic network analysis, and central genes were highlighted. With the central genes, we hypothesized that vascular endothelial microenvironment, VEGF pathway and inflammation may be the possible mechanisms by which LTRAs induce EGPA. The results of this study will have a positive impact on medication safety for patients using LTRAs, which may inform clinical decision-making and the safety-efficacy assessment of LTRA therapy.

## Supporting information

**S1 Table. Checklist of READUS-PV guidelines for reporting disproportionality analyses.**
(DOCX)

**S2 Table. The generic, trade, and former names of leukotriene receptor antagonists.**
(DOCX)

**S3 Table. Formulas for the 4 algorithms in disproportionality analysis.**
(DOCX)

**S4 Table. Related targets of LTRAs from different databases.**
(DOCX)

**S1 Fig. Intersection of EGPA-related genes and LTRAs-related genes.** LTRAs, leukotriene receptor antagonists; EGPA, eosinophilic granulomatosis with polyangiitis.
(TIF)

**S2 Fig. GO analysis of the LTRAs-EGPA genes.** LTRAs, leukotriene receptor antagonists; EGPA, eosinophilic granulomatosis with polyangiitis; GO, Gene Ontology; CC, cellular components; BP, biological processes; MF, and molecular functions; AC−M G protein−CR pathway, adenylate cyclase−modulating G protein−coupled receptor signaling pathway; OA, AO CH−OH GD, NAD or NADP as acceptor, oxidoreductase activity, acting on the CH−OH group of donors, NAD or NADP as acceptor.
(TIF)

## Acknowledgments

We want to acknowledge the participants and investigators of the GEO, FAERS, JADER, CVAR database.

## Author contributions

**Conceptualization:** Cuilv Liang, Yin Zhang.

**Data curation:** Cuilv Liang.

**Formal analysis:** Cuilv Liang.

**Investigation:** Cuilv Liang.

**Methodology:** Cuilv Liang.

**Project administration:** Yin Zhang.

**Software:** Cuilv Liang, Pingping Zhuo.

**Supervision:** Pingping Zhuo, Yin Zhang.

**Writing – original draft:** Cuilv Liang.

**Writing – review & editing:** Cuilv Liang, Pingping Zhuo, Yin Zhang.

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
