## [Decision Letter · Decision Letter 0]

6 Jan 2026

PONE-D-25-55259Leukotriene receptor antagonists and eosinophilic granulomatosis with polyangiitis: a disproportionality analysis from FAERS, JADER, CVAR databases and analysis of the drug-gene interaction networkPLOS One

Dear Dr. Zhang,

Thank you for submitting your manuscript to PLOS ONE. After careful consideration, we feel that it has merit but does not fully meet PLOS ONE’s publication criteria as it currently stands. Therefore, we invite you to submit a revised version of the manuscript that addresses the points raised during the review process.

Please address the reviewers' comments promptly.

We look forward to receiving your revised manuscript.

Kind regards,

Vipula Rasanga Bataduwaarachchi, MD

Academic Editor

PLOS One

2. Please note that PLOS One has specific guidelines on code sharing for submissions in which author-generated code underpins the findings in the manuscript. In these cases, all author-generated code must be made available without restrictions upon publication of the work. Please review our guidelines at https://journals.plos.org/plosone/s/materials-and-software-sharing#loc-sharing-code and ensure that your code is shared in a way that follows best practice and facilitates reproducibility and reuse.

Reviewers' comments:

Reviewer's Responses to Questions

**Comments to the Author**

1. Is the manuscript technically sound, and do the data support the conclusions?

Reviewer #1: Partly

Reviewer #2: Yes

2. Has the statistical analysis been performed appropriately and rigorously? 

Reviewer #1: I Don't Know

Reviewer #2: I Don't Know

3. Have the authors made all data underlying the findings in their manuscript fully available?

Reviewer #1: Yes

Reviewer #2: Yes

4. Is the manuscript presented in an intelligible fashion and written in standard English?

Reviewer #1: Yes

Reviewer #2: Yes

5. Review Comments to the Author

Reviewer #1: 1- rewrite the section Network analysis of LTRAs-EGPA gene interactions?

2- why montelukast remained significantly correlated with EGPA regardless of concomitant corticosteroids use or exclusion?

3- unclear in Fig. 2 presented the characteristics of patients experiencing EGPA re-imaging in high resolutions

Reviewer #2: I appreciate the authors for their time and effort in preparing this manuscript. The study presents a compelling analysis of rare LTRAs-associated adverse events (EPGA) by meticulously analyzing multiple databases from different countries and employing a variety of disproportionality analyses, subgroup analyses, co-medication assessments, and gene-network analyses. However, I have several suggestions that may further strengthen the manuscript and improve its clarity and readability. I believe after considering these points and remarks, this paper can provide valuable information for clinicians and regulatory authorities.

1. The current format of the tables is prone to confusion therefore, I strongly advise reformatting tables. Each table should have clear rows with aligned columns for each entry. Also, if a table contains multiple sections i.e., table 2 and 3, separate each section with proper lines and delineated headings.

2. While SRSs databases contain an extensive amount of information regarding drug use and related adverse events, they have important limitations. Firstly, they don’t show the exact records of drugs dosage changes/tappers or past drug history in this particular study, steroids and disease severity/comorbidities. Secondly, these types of databases are based on the voluntary reports of specific adverse events. Due to these facts, I recommend to clarify whether the authors have already considered this possible bias, when they concluded the existence of significant correlation between montelukast use and EPGA emergence. Since, the unmasking result could be due to steroid tapering or dose changes or patients’ other comorbidities and not solely LTRAs administration.

3. LTRAs are sometimes prescribed for the patients with steroids intolerance or patients with comorbid allergies, which may lead to a false drug-adverse event correlation unrelated to causality. Nevertheless, the authors did not address these potential sources of bias in the manuscript. I advise the authors to clarify if this possible bias have been considered.

4. LTRAs represent a class of drugs and should not be listed next to their individual agents such as montelukast, pranlukast, and zafirlukast. Thus, I advise to use the phrasing “LTRAs, including montelukast, …” or use parentheses “LTRAs (e.g., montelukast, etc,)” consistently throughout the manuscript. (Ref. lines: 29,329,427).

5. Consider replacing “pharmacogenetic network analysis” with “network pharmacology” or “bioinformatics-based network analysis” throughout the manuscript. Because your study only identifies some genes using predicted drug target (from SwissTargetPrediction, CTD, PharmMapper, etc.,), DEG, protein-protein interaction networks via cytoplasmic antibodies and pathway enrichment. These methods are not pharmacogenetic per se, hence, these should not be interchangeably used with pharmacogenetics. Pharmacogenetics include examination of patient genotype that affect drug response, SNPs, inherited risk variations, polymorphisms, etc., which your study lacks such analysis.

6. As per the manuscript the LTRAs selection for this study are montelukast, zafirlukast, pranlukast and ibudilast. However, there is no data provided for ibudilast in the entire manuscript. I recommend to briefly explain why it was excluded from the study.

7. In order to make the paper more comprehensive, it is recommended to add a sentence to explain what are the other indications apart from asthma, specifically when such a definitive statement is mentioned that “the other indications bear a higher signal strength than asthma” (Ref. line 239).

8. The sentence” …. which offers substantial backing for clinical medication choices” seems vague and informal. Therefore, I suggest paraphrasing it to for instance, “…. Which may inform clinical decision-makers and safety-efficacy assessment for LTRAs therapy.” (Ref. line 433).

9. There are some discrepancies/inconsistencies in the manuscript.

a) Line 192, the authors mentioned 949 cases involved montelukast, but later in line 206, it is mentioned “among 415 cases of montelukast”. Is his number a subset from other database or the authors intention is “Among 949 montelukast cases, 415 cases were observed with severe clinical outcomes.” Please clarify these statistics.

b) Some parts in Table 3 represents redundancy. The digits for prednisone in “Concomitant drugs” section is repeated for fluticasone in the “Removal of cases with concomitant drugs” section. I advise to re-check calculations for the fluticasone and prednisone and correct the numbers.

c) In the Results section (3.3 TTO), the authors mentioned that the median TTO of EGPA associated with montelukast was 244 days, but later in the discussion it was stated “median TTO for montelukast is 284 days” (Ref. line 377). Please clarify which one is a correct digit.

d) Line 142, “A positive signal was defined as meeting all four methodological criteria”, however, you have mentioned five criteria.

10. There are some minor Typo/grammar errors.

a) Line 175, “…… (DEGs) in the row dates of GSE…”. Replace date with data.

b) Line 339, the sentence “…. increasing attention has been directed toward determining whether a genuine correlation between LTRAs and EGPA” seems missing a verb. Suggestion; “…. increasing attention has been directed toward determining whether a genuine correlation exist between LTRAs and EGPA.”

c) Ensure consistent capitalization in the table captions. (Ref. line 222, (h)).

d) Line 419, replace microenvironmental with microenvironment.

e) Line 426, replace EGAF with EGPA.

f) It is more appropriate to use comma between multiple factors rather than slash since throughout the manuscript, comma has been used. (Ref. line 148).

6. PLOS authors have the option to publish the peer review history of their article (what does this mean? ). If published, this will include your full peer review and any attached files.

**Do you want your identity to be public for this peer review?** For information about this choice, including consent withdrawal, please see our Privacy Policy .

Reviewer #1: No

Reviewer #2: **Yes:** Dr. Arezou Ahmadi R.A

---

## [Author Response · Author response to Decision Letter 1]

27 Jan 2026

Dear reviewers and editors,

On behalf of all the contributing authors, I would like to express our sincere appreciation of your letter and reviewers’ constructive comments. These comments are all valuable and helpful for improving our article. We have adequately addressed or fully implemented the comments and suggestions from all reviewers involved in the review of the original manuscript. Point-by-point responses to the reviewers are listed below this letter.

Manuscript ID: PONE-D-25-55259

Title: Leukotriene receptor antagonists and eosinophilic granulomatosis with polyangiitis: a disproportionality analysis from FAERS, JADER, CVAR databases integrated with network pharmacology

Reviewer #1:

1. rewrite the section Network analysis of LTRAs-EGPA gene interactions?

Response: Thank you again for your positive comments and valuable suggestions. According to your suggestions, we have made corresponding modifications.

Title: Leukotriene receptor antagonists and eosinophilic granulomatosis with polyangiitis: a disproportionality analysis from FAERS, JADER, CVAR databases integrated with network pharmacology

Introduction: Network pharmacology is a rapidly advancing technology that integrates computer science and bioinformatics to explore drug-target interactions and signaling pathways. We employed network pharmacology to elucidate the mechanism of action of LTRAs in EGPA.

2. Materials and methods:

2.5 Network pharmacology analysis: LTRAs-related EGPA network construction

The data required for the comprehensive analysis was obtained from online database platforms [32]. For LTRAs-related targets, including montelukast, zafirlukast, pranlukast, and ibudilast, we searched the SwissTargetPrediction database (https://www.swisstargetprediction.ch/), the Comparative Toxicogenomics database (CTD; https://ctdbase.org/), the Targetnet database (http://targetnet.scbdd.com/home/index/), and the PharmMapper database (https://www.lilab-ecust.cn/pharmmapper/index.html) to predict possible targets. The EGPA-related genes were identified from differentially expressed genes (DEGs) in the row data of GSE144302 (https://www.ncbi.nlm.nih.gov/geo/).

The drug-target list and disease-DEG list were submitted to VENN diagram for obtaining overlapping targets (LTRAs-EGPA genes). The overlapping targets of LTRAs and EGPA were the potential targets that could regulate EGPA. The overlapping targets were investigated by the STRING website (https://string-db.org/), with a focus on Homo sapiens species with a 0.4 confidence threshold. Then, Cytoscape software was employed to visualize the protein-protein interaction network. In this network, biological entities (LTRAs-EGPA-related genes) were depicted as nodes, and the interactions between them were represented as edges. The molecular complex detection (MCODE) algorithm was applied to screen for key cluster [33].

The LTRAs-EGPA genes were analyzed by Gene Ontology (GO) and Kyoto Encyclopedia of Genes and Genomes (KEGG) for functional enrichment [34]. Through GO analysis of these genes, we identified three categories of cellular components, biological processes, and molecular functions to investigate the biological characteristics of LTRAs-associated EGPA genes. Meanwhile, KEGG enrichment analysis was performed to predict potential signaling pathways that may be implicated in LTRAs-induced EGPA.

3. Results:

3.4 LTRAs-related EGFA network construction

After deduplication of the databases, we identified 553 target genes linked to LTRAs, along with 3,555 genes linked to EGPA (Supplementary Table 4). By intersecting these genes, we separated 81 interacting target genes that represented the intersection of LTRAs target genes and EGPA-related genes (Supplementary Fig. 1). We conducted protein-protein interaction predictions for these intersecting genes via the String database (Fig. 5a). Upon the MCODE algorithm, a key cluster of the interaction was identified, including SRC, PTGS2, EDN1, HMOX1, KDR, and OCLN, revealing their centrality within the network (Fig. 5b).

KEGG analysis of LTRAs-EGPA genes was shown in Fig. 5c. We focused on the top 20 pathways for comprehensive mapping. The analysis revealed that genes interacting with LTRAs-EGPA were enriched in different pathways, particularly VEGF signaling pathway, renin secretion, and drug metabolism. These findings suggested that LTRAs may be involved in the development of EGPA by regulating these pathways. The GO analysis results of the LTRAs-EGPA genes were presented in Supplementary Fig. 2. The biological process category of GO exhibited enrichment in vascular processes in circulatory system and muscular systems. The cellular component category was enriched in membrane raft and membrane microdomain. The molecular function category showed enrichment in amide binding oxidoreductase activity and acting on the CH-OH group of donors, NAD or NADP as acceptor.

Fig. 5 Results of gene interaction network analysis of LTRAs-associated EGPA

(a)Protein-protein interaction network of LTRAs-associated EGPA genes presented by Cytoscape.

(b)Key cluster genes based on the MCODE algorithm.

(c) KEGG pathway enrichment analysis of LTRAs-associated EGPA genes

References

32. Yao W, Huo J, Ji J, Liu K, Tao P Elucidating the role of gut microbiota metabolites in diabetes by employing network pharmacology. Mol Med. 2024; 30(1): 263. https://doi.org/10.1186/s10020-024-01033-0.

33. Xie R, Li B, Jia L, Li Y Identification of core genes and pathways in melanoma metastasis via bioinformatics analysis. Int J Mol Sci. 2022; 23(2). https://doi.org/10.3390/ijms23020794

34. Yu G, Wang LG, Han Y, He QY Clusterprofiler: An r package for comparing biological themes among gene clusters. Omics. 2012; 16(5): 284-287. https://doi.org/10.1089/omi.2011.0118

2.why montelukast remained significantly correlated with EGPA regardless of concomitant corticosteroids use or exclusion?

Response: Thank you again for your positive comments. There have been some case reports supporting the absence of a direct causal relationship between montelukast and EGPA, as EGPA was observed in asthma patients only after reducing or discontinuing oral corticosteroids [17-19]. To discern the effects of corticosteroid co-administration on EGPA after taking LTRAs, we used four methods to perform subgroup analyses by co-medication analysis of corticosteroid use. In our study, montelukast remained significantly correlated with EGPA regardless of concomitant corticosteroid use or exclusion.

References

17.Harrold LR, Patterson MK, Andrade SE, Dube T, Go AS, Buist AS, et al. Asthma drug use and the development of churg-strauss syndrome (css). Pharmacoepidemiol Drug Saf. 2007; 16(6): 620-626. https://doi.org/10.1002/pds.1353

18. Wechsler ME, Finn D, Gunawardena D, Westlake R, Barker A, Haranath SP, et al. Churg-strauss syndrome in patients receiving montelukast as treatment for asthma. Chest. 2000; 117(3): 708-713. https://doi.org/10.1378/chest.117.3.708

19. Cuchacovich R, Justiniano M, Espinoza LR Churg-strauss syndrome associated with leukotriene receptor antagonists (ltra). Clin Rheumatol. 2007; 26(10): 1769-1771. https://doi.org/10.1007/s10067-006-0510-0

3. unclear in Fig. 2 presented the characteristics of patients experiencing EGPA re-imaging in high resolutions

Response: Thank you for your comment regarding Fig. 2. We apologize for any confusion caused by the wording. Fig. 2 presents the basic information of LTRAs associated with EGPA toxicity through bar charts, line plots, and pie charts. It does not include any imaging data or high-resolution re-imaging. The term "re-imaging" does not apply to this figure, as it contains only statistical data visualizations. To avoid any further misunderstanding, we have added a clarifying note to the figure legend:

Note: Figure 2 contains only statistical data visualizations and does not include any patient imaging or re-imaging data.

Fig. 2 Basic information of LTRAs associated with EGPA toxicity. (a) The number of annual adverse reaction reports. (b) The gender distribution of the patients. (c) The age distribution of the patients. (d) The weight distribution of the patients. (e) The occupational distribution of the reporter. (f) The top 8 countries with the largest number of montelukast-related EGPA reports. (g) The outcome distribution of montelukast-related EGPA in patients. (h) The outcome distribution of pranlukast-related EGPA in patients. Note: Fig. 2 contains only statistical data visualizations and does not include any patient imaging or re-imaging data.

LTRAs, leukotriene receptor antagonists; EGPA, eosinophilic granulomatosis with polyangiitis; FAERS, Food and Drug Administration Adverse Event Reporting System; JADER, Japanese Adverse Drug Event Reporting; CVAR, Canadian Vigilance Adverse Reaction; NHP, non-health professional; OHP, other health professional.

Reviewer #2:

1. The current format of the tables is prone to confusion therefore, I strongly advise reformatting tables. Each table should have clear rows with aligned columns for each entry. Also, if a table contains multiple sections i.e., table 2 and 3, separate each section with proper lines and delineated headings.

Response: Thank you for this important stylistic suggestion. We have completely re-formatted all tables according to your suggestion.

(1) Aligned columns: every numerical cell is now right-aligned; text cells are left-aligned.

(2) Clear sectioning: Tables 2 and 3 contain multiple sub-analyses; each sub-section (Gender, Age, Indication, Concomitant drugs …) is separated by a solid horizontal line and preceded by a sub-heading (e.g., A. Gender, B. Age, C. Indication).

Font and spacing:10 pt Times New Roman, 2-line spacing; no vertical borders.

We added this point in the revised manuscript and the detailed revision can be found in Table 1-4.

2. While SRSs databases contain an extensive amount of information regarding drug use and related adverse events, they have important limitations. Firstly, they don’t show the exact records of drugs dosage changes/tappers or past drug history in this particular study, steroids and disease severity/comorbidities. Secondly, these types of databases are based on the voluntary reports of specific adverse events. Due to these facts, I recommend to clarify whether the authors have already considered this possible bias, when they concluded the existence of significant correlation between montelukast use and EPGA emergence. Since, the unmasking result could be due to steroid tapering or dose changes or patients’ other comorbidities and not solely LTRAs administration.

Response: Thank you again for your positive comments and valuable suggestions. We fully acknowledge these limitations. We have now explicitly highlighted the inherent limitations of these SRSs and described the analytical remedies we employed. Although three databases encompass millions of AE reports, they are SRSs with inherent limitations. First, these systems do not provide an exact record of drug dosage changes or tapering regimens, prior medication history, disease severity, or detailed comorbidity data in a structured or comprehensive manner. Consequently, we were unable to adjust for time-varying corticosteroid doses or perform analyses corrected for disease severity. Second, the anonymous, voluntary nature of SRSs makes them susceptible to under-reporting, duplicate reports, incomplete data, and reporting bias. To mitigate these limitations, the following measures were taken:

(1) Three large-sample databases and four data mining algorithms were utilized for cross-validation.

(2) Disproportionality analyses were restricted to AE reports in which LTRAs were coded as the "principal suspect" in the FAERS database or as a "suspect" drug in the JADER and CVAR databases, to minimize dilution effects from concomitant medications.

(3) Four methods were used to conduct a co-medication analysis of corticosteroid use. Montelukast remained significantly associated with EGPA regardless of concomitant corticosteroid use or exclusion, thereby minimizing confounding by steroid therapy.

(4) Analyses stratified by age, gender, and indication were conducted. The signal strength remained consistent across all strata, indicating that the signal was not driven by a single demographic characteristic or indication cluster.

Despite these sensitivity analyses and subgroup analyses, residual confounding cannot be excluded due to the lack of individual-level data on corticosteroid tapering and comprehensive comorbidity information. Our results support the presence of a signal consistent with a potential causal role for LTRAs. Disproportionality analysis also can only assess signal strength. Therefore, the findings of this study cannot establish causality or quantify the independent contribution of LTRAs relative to corticosteroid dose reduction or other comorbidities. Further validation is required through longitudinal prescription databases that include dosage and diagnostic timelines or prospective pharmacovigilance cohort studies.

We added this point to the revised manuscript and the detailed revision can be found in page 22 (line 421-424), page 23 (line 425-437).

3. LTRAs are sometimes prescribed for the patients with steroids intolerance or patients with comorbid allergies, which may lead to a false drug-adverse event correlation unrelated to causality. Nevertheless, the authors did not address these potential sources of bias in the manuscript. I advise the authors to clarify if this possible bias have been considered.

Response: Thank you again for your positive comments and valuable suggestions. As previously noted, LTRAs are sometimes prescribed for patients with steroid intolerance or comorbid allergic conditions, which may lead to spurious drug-event associations. Given the inherent limitations of SRS data, we cannot obtain exact information on steroid intolerance and comorbid allergic conditions in patients in a structured or comprehensive manner.

But we attempted to detect and mitigate such biases as follows:

(1) Four methods were used to perform a co-medication analysis of corticosteroid use. Montelukast remained significantly associated with EGPA regardless of concomitant corticosteroid use or exclusion, indicating that the signal is not solely attributable to steroid therapy.

(2) Using the FAERS database as an example, the INDI table (which records only the indication for the currently reported drug and does not systematically list all patient comorbidities or prior diseases, yet still offers some reference value) showed that only 8.16% of patients had documented comorbid allergic conditions, suggesting a relatively minor impact on our study.

Despite our attempts to mitigate these biases, their source cannot be fully eliminated. Consequently, the observed association may still overestimate the true causal effect of LTRAs on EGPA. Prospective studies utilizing linked electronic health records or claims databases capable of capturing detailed contraindications and prior comorbidities are warranted to quantify the independent contribution of LTRA exposure.

We added this point to the revised manuscript and the detailed revision can be found in page 22 (line 421-424), page 23 (line 425-437).

4. LTRAs represent a class of drugs and should not be listed next to their individual agents such as montelukast, pranlukast, and zafirlukast. Thus, I advise to use the phrasing “LTRAs, including montelukast, …” or use parentheses “LTRAs (e.g., montelukast, etc,)” consistently throughout the manuscript. (Ref. lines: 29,329,427).

Response: Thank you again for your positive comments and valuable suggestions. According to your suggestions, we have made corresponding modifications. All instances of “LTRAs/montelukast/pranlukast/zafirlukast” or class-agent listing have been revised throughout the manuscript. We added this point to the revised manuscript and the detailed revision can be found in page 2 (line 28, 37), page 11 (line 238-241), page 16 (line 292-297), page 17 (line 298-301,305-306), page 19 (line 342-344), page 19 (line 447-450).

5. Consider replacing “pharmacogenetic network analysis” with “network pharmacology” or “bioinformatics-based network analysis” throughout the manuscript. Because your

---

## [Editor Report · Decision Letter 1]

1 Feb 2026

Leukotriene receptor antagonists and eosinophilic granulomatosis with polyangiitis: a disproportionality analysis from FAERS, JADER, CVAR databases integrated with network pharmacology

PONE-D-25-55259R1

Dear Dr Zhang,

We’re pleased to inform you that your manuscript has been judged scientifically suitable for publication and will be formally accepted for publication once it meets all outstanding technical requirements.

Within one week, you’ll receive an email detailing the required amendments. When these have been addressed, you’ll receive a formal acceptance letter ,and your manuscript will be scheduled for publication.

An invoice will be issued upon formal acceptance of your article. Please note that if your institution has a publishing partnership with PLOS and your article meets the relevant criteria, all or part of your publication costs will be covered. Please make sure your user information is up-to-date by logging into Editorial Manager at Editorial Manager®  and clicking the ‘Update My Information' link at the top of the page. For questions related to billing, please contact billing support .

If your institution or institutions have a press office, please notify them about your upcoming paper to help maximise its impact. If they’ll be preparing press materials, please inform our press team as soon as possible -- no later than 48 hours after receiving the formal acceptance. Your manuscript will remain under strict press embargo until 2 pm Eastern Time on the date of publication. For more information, please contact onepress@plos.org.

Kind regards,

Vipula Rasanga Bataduwaarachchi, MD

Academic Editor

PLOS One

---

## [Editor Report · Acceptance letter]

PONE-D-25-55259R1

PLOS One

Dear Dr. Zhang,

I'm pleased to inform you that your manuscript has been deemed suitable for publication in PLOS One. Congratulations! Your manuscript is now being handed over to our production team.

Kind regards,

on behalf of

Dr. Vipula Rasanga Bataduwaarachchi

Academic Editor

PLOS One